# Dose- and time-dependent changes in viability and IL-6, CXCL8 and CCL2 production by HaCaT-cells exposed to cobalt. Effects of high and low calcium growth conditions

Maria Klasson[1,2,3]*, Magnus Lindberg[1,2,4], Eva Särndahl[1,2], Håkan Westberg[1,2,3], Ing-Liss Bryngelsson[3], Kedeye Tuerxun[1,2], Alexander Persson[1,2]

**1** Department of Medical Sciences, School of Medicine and Health, Örebro University, Örebro, Sweden,
**2** Inflammatory Response and Infection Susceptibility Centre (iRiSC), Örebro University, Örebro, Sweden,
**3** Department of Occupational and Environmental Medicine, University Hospital Örebro, Örebro, Sweden,
**4** Department of Dermatology, University Hospital Örebro, Örebro, Sweden

* maria.klasson@regionorebrolan.se

## Abstract

### Background

Sensitization requires exposure to an allergen with subsequent production of a "danger "signal. In the skin, keratinocytes are the main producers of these signals.

### Objective

To compare dose- and time-effects of cobalt on the viability of and cytokine release from HaCaT cells cultured at low or high calcium.

### Method

To model two separate states of differentiation of keratinocytes, HaCaT cells were cultured under low or high calcium conditions. HaCaT were exposed to different concentrations of cobalt chloride (10 µm to 5 mM) over time (30 minutes– 48 hours). Cell viability was measured with the Cell-Titer Blue Viability assay. Cytokine production was measured using a bead-based immunoassay and flow cytometry. Gene expression was quantified using qPCR. Data was analyzed by ANOVA and linear mixed model.

### Results

Viability of the cells was dose- and time-dependent. A linear mixed statistical model showed that cobalt exposure induces increase in IL-6, CXCL8 and CCL2 production over time and whereas increase of IL-6 and a decrease of CCL2 was associated with increasing cobalt chloride concentrations. When comparing the cells incubated under high and low calcium conditions, the more differentiated cells in the high concentration were found to exert a stronger response in terms of IL-6 release.

**Funding:** This project was funded by the Synergy15-grant from the Knowledge Foundation (ES), grant number 20160044, URL: www.kks.se, Grant Hudfonden (ML), grant number 2499/2016:1, URL: www.hudfonden.se, and the ALF funding Region Örebro County (MK), grant number OLL-888151, URL: www.regionorebrolan.se. The funders had no role in study design, data collection and analysis, decision to publish, or preparation of the manuscript.

**Competing interests:** The authors have declared that no competing interests exist.

## Conclusions

Our data suggest that cobalt chloride triggered an alarm system in HaCaT cells, and proinflammatory cytokines/chemokines were secreted in a dose- and time-dependent manner. When high and low calcium incubations were compared, the difference was seen only for IL-6. These findings indicate that the effect of cobalt chloride on cell toxicity occurs throughout the living epidermis.

## Introduction

The prevalence of contact allergy to cobalt among patch tested patients in Sweden was shown to be 4,5% in 2009 [1]. Cobalt can induce allergic contact dermatitis both by direct contact with the skin and by airborne particle exposure [2–4]. Skin exposure to cobalt is known to occur in the hard metal production industry [2], from jewelry, coins, implants and various alloys containing cobalt [3, 5].

Cobalt, both in the form of powder and as solid material metal, ionizes in the presence of artificial sweat [6–8]. When ionized, *in vitro* studies have demonstrated that cobalt can penetrate the skin barrier [7], and we have previously shown this to contribute to systemic uptake of cobalt to the blood [9]. Keratinocytes have the ability to respond to stimuli from our environment and exert inflammatory mediation in the skin through the production of several pro-inflammatory cytokines as a response to various danger signals [10, 11]. These soluble immune-modulating factors from keratinocytes have been shown to be involved in delayed-type hypersensitivity reactions i.e., allergic contact dermatitis making keratinocytes an important component in the initiation of these maladies [12].

Cobalt have shown to be cytotoxic to human keratinocytes [13–16] and the model often used for keratinocyte cobalt exposure, immortalized HaCaT cells, show decrease cell viability in a dose-dependent manner following cobalt chloride ($CoCl_2$) exposure. In addition, $CoCl_2$ exposure results in a dose-dependent increase of intracellular reactive oxygen species and secretion of TNF, IL-1β, IL-6 and CXCL8 [15, 17–19].

Keratinocytes are present in the epidermis at different levels of differentiation, with the basal layers comprising the majority of undifferentiated cells. In contrast, cells in the upper part of the epidermis are increasingly differentiated. Although the effects of cobalt exposure in vitro have been studied, the impact of cobalt exposure at different depths of skin i.e., various levels of differentiation of the keratinocytes is unknown.

The primary aim of this study was to describe dose- and time-dependent cytotoxicity and cytokine/chemokine release from keratinocytes following exposure to cobalt. A secondary aim was to investigate potential differences in cellular responses to cobalt exposure depending on the level of cellular differentiation. To approach this, we used HaCaT cells in a model where cell culture under low calcium conditions represents low differentiated keratinocytes in the basal layer, and high calcium conditions represent more differentiated cells in the epidermis.

## Materials and methods

In this study, the human immortalized cell line HaCaT was used as a proxy model for keratinocytes in human skin. Two different calcium concentrations during cell culture were used to simulate increasing HaCaT cell-differentiation to resemble basally located cells and suprabasally located cells in the epidermis, respectively. We used the metabolic activity as a measure

for cell cytotoxicity, immunoassay to determine the release of cytokines/chemokines and PCR to determine the gene expression levels of the cytokines in HaCaT cells following $CoCl_2$ exposure.

## Materials

Dulbecco´s modified Eagle´s medium (DMEM) with and without calcium, fetal bovine serum (FBS), glucose, gentamicin, L-Glutamine, trypsin-EDTA (0.25%), Dulbecco's phosphate-buffered saline (DPBS) were all purchased from Thermo Fisher Scientific Inc, Waltham MA. Calcium levels in DMEM were adjusted to 0.15 mM and 1.8 mM. Cell Titer-Blue Cell Viability Assay was from Promega, Madison, WI. $CoCl_2$ 0.1 M solution was purchased from Sigma-Aldrich, St. Louis, MI. The cells used in this study were the immortalized human keratinocyte cell line HaCaT [16] and generously provided by Anita Koskela von Sydow and Mikael Ivarsson (Örebro University, 701 82 Örebro, Sweden).

## Cell culture

HaCaT cells were maintained in DMEM containing 0.15 mM calcium and supplemented with 10% FBS, 4 mM L-glutamine and 1 µg/mL gentamicin at 37˚C and 5% $CO_2$ and the cells were passaged with trypsin when reaching an estimated 70% confluency. Before experiments, cells were cultured in DMEM containing 1.8 mM calcium for seven days to induce differentiation or kept in medium containing 0.15 mM calcium for the undifferentiated phenotype. The cell culture was regularly tested for mycoplasma contamination using the MycoAlert™ Mycoplasma Detection Kit according to manufacturer's instructions (Lonza, Basel, Switzerland).

## Cell viability assay

Cells were trypsinized and seeded in 96- well plates (Greiner Bio-One GmbH, Kremsmünster, Austria) at day six and incubated for another 24 hours at a density of $9x10^3$ cells/well in a volume of 100 µL medium. $CoCl_2$ diluted in DMEM was added to the cells at concentrations of 5 mM, 2.5 mM, 1 mM, 500 µM, 250 µM, 100 µM, 10 µM and incubated for 3h, 6h, 12h, 24h or 48h. For cell viability quantifications, cell culture media was substituted for 20 µl Cell-Titer Blue reagent for 2.5 hours and analyzed with Fluo Star Optima (BMG LABTECH GmbH, Ortenberg, Germany). The experiments were performed four times, with two replicates for each time point and concentration. To investigate how cobalt exposure affected cell vitality over time, cell cultures were exposed to 100 µM $CoCl_2$, and cultured for 14 days with subsequent cell viability assessment.

## Cytokine and chemokine measurements

Cells were seeded in 24-well plates at day six at a density of $4x10^5$ cells/well in a volume of 500 µL medium for 24 hours before the start of experiments. Cells were exposed to 100 µM, 500 µM or 1000 µM of $CoCl_2$ to investigate the inflammatory response in HaCaT cells following exposure to $CoCl_2$. Supernatants were collected after 30 minutes, 24 hours and 48 hours. Exposure time 30 minutes function as a model control time. Supernatants were centrifuged and aliquoted and stored at—80˚C until analysis. Experiments were performed five times with three technical replicates.

Cytokine/chemokine production was measured using the immunoassay LEGENDplex Human Inflammation Panel from BioLegend (San Diego, CA) analyzing IFN-α, IFN-γ, TNF, CCL2, IL-6, CXCL8, IL-10, IL-12p70 IL-17A, and IL-23 according to manufacturer instructions using an Accuri C6 (Becton Dickinson, San Jose, CA).

## Gene expression analysis

Cells were seeded in 48 well plates at day six at a density of $1 \times 10^5$ cells/well in a volume of 200 μL medium for 24 hours before the start of experiments. $CoCl_2$ was diluted in DMEM to 500 μM and 1000 μM and added to the wells at the start of exposure.

Following 24 and 48 hours of cobalt exposure cells were lysed in RLT lysis buffer and total RNA was extracted using RNeasy Mini Kit (Qiagen, Hilden, Germany) according to the manufacturer's instructions. The RNA quality was measured with 2100 Bionalyzer (Agilent, Santa Clara, CA) and quantified using NanoDrop 2000 (Thermo Fisher Scientific, Waltham, MA). cDNA was synthesized in an 80 μl reaction containing 1.2 μg RNA, using the High Capacity cDNA Reverse Transcription Kit (Applied Biosystems, Foster City, CA) in a LifePro Thermal Cycler (Bioer, Hangzhou, China).

## Real-time PCR

Real-time PCR was performed in a Quantstudio 7 Flex Real-Time PCR system on 1 μl sample cDNA in a final reaction volume of 10 μL using TaqMan Fast Universal PCR Master Mix and TaqMan Gene Expression Assays for *IL6* (Hs00174131), *CXCL8* (Hs00174103) and *CCL2* (Hs00234140) all from Applied Biosystems (Foster City, CA), according to the manufacturer's instructions. Water was included as a negative control, and six random samples were included as no reverse transcriptase control (NRT) in every run to assess the amount of gDNA contamination. Liquid handling into a 384-well plate was performed by a PIRO pipetting robot (Dornier, Lindau, Germany). *TBP* and *PPIB* were determined by using the NormFinder R package (MOMA, Aarhus University Hospital, Denmark) as reference genes for normalization among a total of four candidate reference genes. For quantification, a six-point serially four-fold diluted calibration curve was developed from peripheral blood mononuclear cells stimulated by 1 μg/ml lipopolysaccharide. All samples were amplified in duplicate, and the mean quantity values were obtained for further data analysis. An acceptable coefficient of variation (CV) between duplicates was set to <15%, and cycle threshold (CT) cut-off value was set to 35. All reactions had an efficiency between 90% and 110%, which corresponds to a slope between -3.58 and -3.10. All data were collected via QuantStudio™ Real-Time PCR software (Applied Biosystems, Foster City, CA).

## Statistical methods

The cell viability results are expressed as fluorescence signal of exposed cells compared to unexposed control cells (the latter regarded as 100% viability). Results are presented as a median value for each $CoCl_2$ concentration and time point and for the long-term viability experiments, also range (minimum and maximum values) is shown.

Results from immunoassays are presented with each point representing concentration data from one measurement (three technical replicates) and the median value. Data from gene expression analysis are presented as a median value of the relative mRNA levels and range (min-max). ANOVA was used to validate whether there are differences between the controls and respective test parameter in the experimental set up. The lowest exposure cobalt concentration, lowest exposure time, and low calcium concentration are set as the reference category in the analysis. The results of the ANOVA analysis are presented as logarithmic mean value and 95% confidence intervals. A linear mixed model was further developed to describe the relationship between the different exposure times, $CoCl_2$ concentrations, and high- and low calcium conditions used in the cells models. Our experimental design with repeated measurements and the use of a mixed model made it possible to consider variability within the same experiment and variability between the repeated experiments. In the mixed linear model, as

we changed one input variable at a time and kept the other variables constant, we could reveal correlations for each of the exposure time, $CoCl_2$ concentration, and high and low calcium culture conditions. The estimates (β) of the fixed effects from the model allowed us to identify factors affecting the production of cytokines/chemokines and gene expression.

Since the distribution of cytokines/chemokines and genes was log normally distributed, data were log-transformed. The equation of the final mixed model is:

$$Y = \ln(X) = \mu + \beta_1[\text{EXPOSURE TIME}] + \beta_2[\text{COBALT CHLORIDE CONCENTRATION}]$$
$$+ \beta_3[\text{HIGH/LOW CALCIUM INCUBATION}] + \varepsilon$$

$Y = \ln(X)$; X and Y are the measured and log-transformed cytokine/chemokine concentrations and normalized gene expression, respectively.
$\mu$ = the overall average IL-6, CXCL8, CCL2 concentration and *IL6* mRNA, *CXCL8* mRNA, *CCL2* mRNA normalized expression on the log-scale.
$\beta_1$ = the fixed effect of the exposure time (30 min reference, 24 and 48 hours for cytokines/chemokines and 24 hours reference and 48 hours for the gene normalized expression).
$\beta_2$ = $CoCl_2$ concentration (0 μM reference, 100 μM, 500 μM, 1000 μM respectively for cytokines/chemokines and 0μM reference, 500 μM, 1000 μM for gene expression).
$\beta_3$ = high and low calcium incubation (1.8 mM and 0.15 mM respectively).
$\varepsilon$ = residual.

The lowest exposure cobalt concentration, lowest exposure time, and low calcium concentration are serving as the reference category. The results of the linear mixed model analysis are presented as antilogarithmic β-values (odds ratio) and 95% confidence intervals. The linear model analysis also adjusts for within- and between individual variability, thus improving any linear model fit. The statistical significance threshold was $p < 0.05$ in the ANOVA and in the linear mixed model. Two outliers, representing apparent erroneous handling in immunoassay, were excluded from the analyses.

Testing of normality was performed using Kolmorogov-Smirnov, IBM SPSS Statistics 22. Cell viability analyses were performed using GraphPad Prism 5.03 software (San Diego, CA), and data from immunoassays and gene expression data were analyzed using ANOVA and linear mixed model analyses with IBM SPSS Statistics 25 (Chicago, IL).

## Results

### Cell viability/toxicity

Cells cultured in low calcium medium were more affected by the $CoCl_2$ exposure than cells in high calcium medium (Fig 1A and 1B). Significant difference between low- and high calcium incubations was seen at six hours for $CoCl_2$ concentrations 250 μM and 500 μM and at 24 hours for 100 μM and 1000 μM $CoCl_2$. The long-term effect of low cobalt exposure on cell viability for 14 days was non-significantly different from control cells (Fig 2). Control cells display a variation in readout between 91–114% over 14 days.

### Cytokine production/inflammatory response

HaCaT cells exposed to $CoCl_2$ for 24 and 48 hours showed an increase in IL-6 production compared to controls (Fig 3). In the ANOVA analyses (Table 1), a significant difference for IL-6 production mean (pg/ml) between the shortest exposure time (30 minutes), serving as control, and 24 hours (2.81, 95% CI 2.29–3.33) and 48 hours (3.08, 95% CI 2.56–3.58) was seen. Difference was also seen between the unexposed cells, 0 μM, and the highest test

## A Low- calcium

## B High-calcium

**Fig 1. Cell viability for HaCaT cells exposed to cobalt chloride under low-calcium and high-calcium growth conditions.** Cobalt chloride exposure concentrations are ranging from 10 μM to 5 mM and exposure times are 3, 6, 12, 24 and 48 hours. The cell viability was measured with Cell Titer Blue viability assay and a dose- and time dependent decrease in viability was seen for low-calcium (A) and high calcium (B) growth conditions.

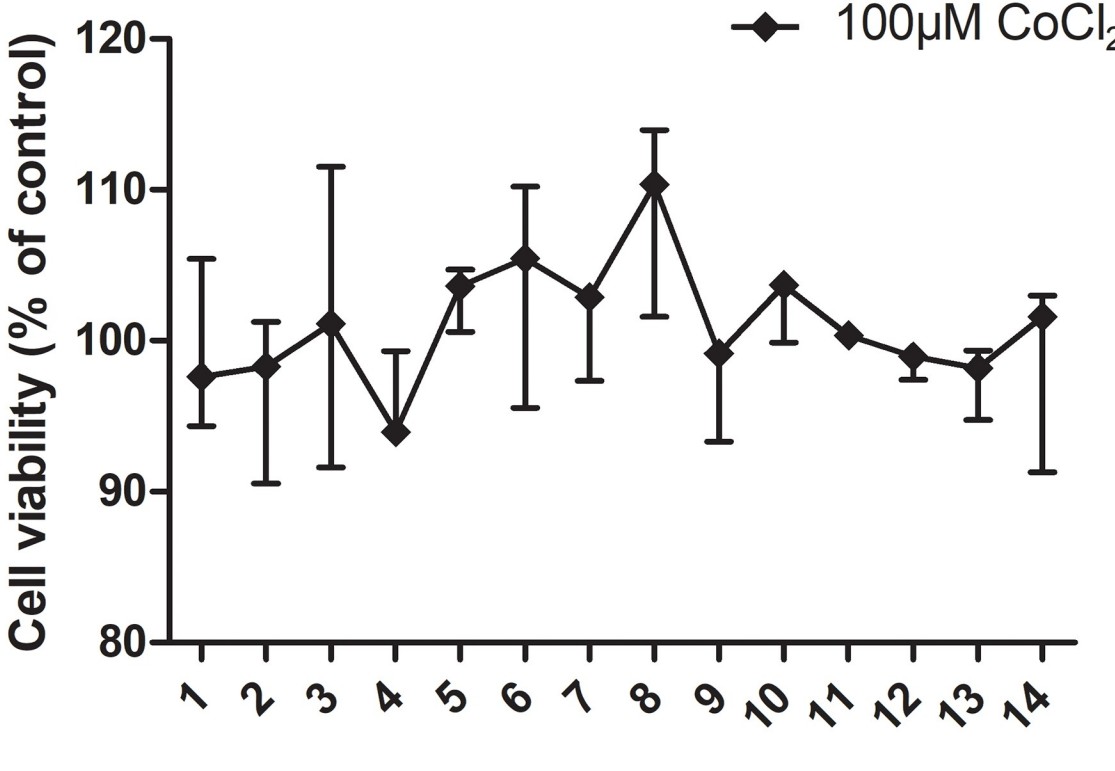

**Fig 2. Cell viability of HaCaT cells cultured in 0.15 mM calcium and exposed to 100μM CoCl₂ for 14 days.** The cell viability is expressed as fluorescence signal of exposed cells compared to unexposed control cells (the latter regarded as 100% viability). The values represent mean and range (minimum and maximum values). The long-term effect of low cobalt exposure on cell viability for 14 days was non-significantly different from control cells.

concentration, 1000 μM (3.24, 95% CI 2.52–3.97) and between the highest and lowest calcium incubation (2.10, 95% CI 1.68–2.53). In the linear mixed model (Table 2), significant increases for IL-6 was determined at 24 and 48 hours (OR 3.93, 95% CI 2.08–7.43 and OR 5.11, CI = 2.7–9.67). CoCl₂ concentration of 100 μM and 500 μM did not affect the IL-6 concentration, whereas 1000 μM induced a significant increase in IL-6 secretion (OR 3.5, 95% CI 1.67–7.32). High calcium levels representing the more differentiated cells, showed significantly higher IL-6 levels than low calcium representing less differentiated cells (OR 2.0, 95% CI = 1.19–3.37).

CXCL8 production mean (pg/ml) in HaCaT (Fig 4) increased over time with significant increase between 30 minutes exposure time and 24 (6.92, 95% CI 6.52–7.31) and 48 hours (7.14, 95% CI 6.67–7.60) (Table 1). Using the linear mixed model an increase at with significant increase at 24 and 48 hours (OR 8.49, 95% CI 4.58–15.73, OR 10.56, 95% CI 5.70–19.57) (Table 2). Non-significant increases of CXCL8 were noted for the different CoCl₂ concentrations. No difference between high and low calcium incubation was found.

Production of CCL2 decreased with increasing CoCl₂ exposure, but showed an overall higher production with increasing time (Fig 5). In Table 1, a significant increase in CCL2 production mean (pg/ml) was seen at 24 and 48 hours (4.25, 95% CI 3.71–4.79 and 4.85, 95% CI 4.33–5.37), also seen while using the mixed linear model (Table 2) at 24 and 48 hours (OR

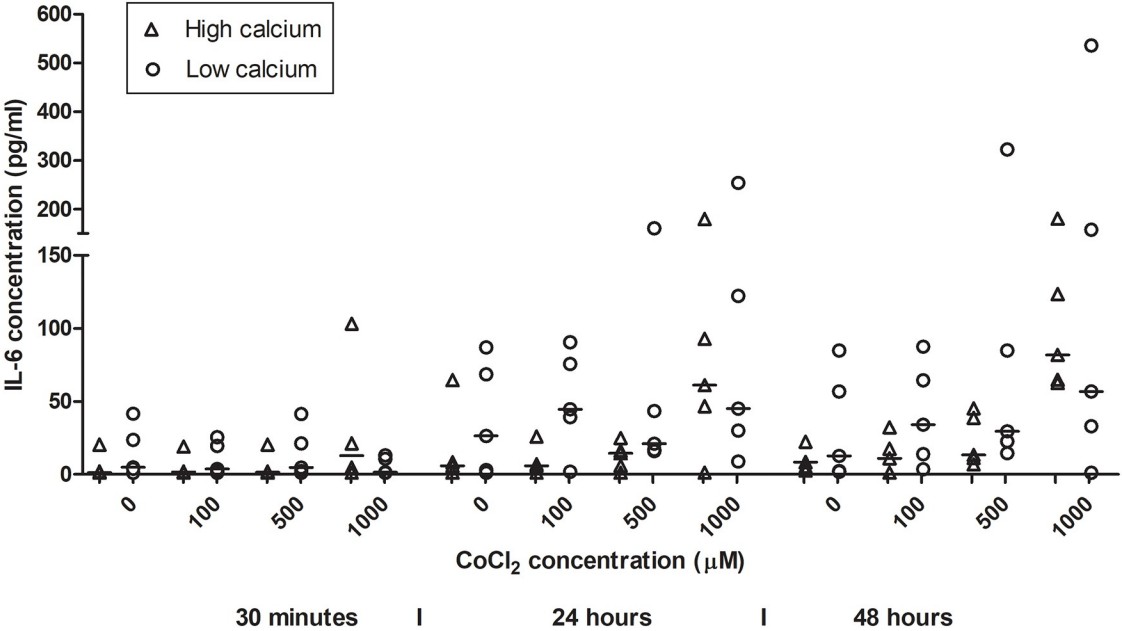

**Fig 3. IL-6 secretion from HaCaT cells after exposure to cobalt chloride.** HaCaT cells were exposed to cobalt chloride concentrations are 100, 500 and 1000 μM and an unexposed control (0 μM). Exposure times are 30 minutes, 24 hours and 48 hours and calcium incubation of HaCaT cells in high (1.8 mM) and low (0.15 mM) calcium concentrations are used. The secretion of IL-6 increased over time and with increasing cobalt chloride concentrations.

5.98, 95% CI 2.88–12.44 and OR 10.95, 95% CI 5.27–22.77). The mixed linear model also revealed a significant decrease in CCL2 production for cobalt concentrations 500 μM and 1000 μM (OR 0.33, 95% CI 0.14–0.76 and OR 0.26, 95% CI 0.11–0.61). No significant difference between high- and low-differentiated cells was detected.

## Cytokines below detection limit

The immunoassay resulted in values below the limit of detection for cytokines IFN-α, IFN-γ, TNF, IL-10, IL-12p70, IL-17A and IL-23 for all the tested time points and $CoCl_2$ concentrations.

## Relative mRNA/gene expression

When gene expression of HaCaT cells exposed to $CoCl_2$ was investigated (Fig 6A–6C) a significant difference in *IL-6*, *CXCL8* and *CCL2* mRNA production, shown as relative expression, was seen (Table 3) when the unexposed cells (0 μM) was compared to 500 μM (-7.22, 95% CI -7.47–(-6.98), -6.64 95% CI -7.03–(-6.26) and -8.38, 95% CI -8.67- (-8.08), respectively) and 1000 μM (-5.20, 95% CI -5.35–(-5.05), -4.88, 95% CI -5.03–(-4.72) and -7.33, 95% CI -8.25, 95% CI -8.55- (-7.96), respectively). Significant difference was also detected when exposure time increased from 24 hours to 48 hours for *CXCL8* (-6-03, 95% CI -6.47–(-5.65)) and *CCL2* (-8.20, 95% CI -8.48-(-7.91)) mRNA. Using the linear mixed model (Table 4), a significant increase in production was seen at 500 μM and 1000 μM for *IL6* mRNA (OR 3.24, 95% CI 1.12–9.34, and OR 23.31, 95% CI 10.05–54.09) and *CXCL8* mRNA (OR 1.83, 95% CI 1.36–2.46, and OR 9.55, 95% CI 6.89–13.24) (Table 2). *CCL2* mRNA production decreased at 500 μM and 1000 μM (OR 0.35, 95% CI 0.26–0.47, and OR 0.42, 95% CI 0.30–0.58). Increment of the exposure time from 24 hours to 48 hours decreased the production of *IL6* mRNA (OR

**Table 1. ANOVA analyses for dose- and time-trends for IL-6, CXCL8 and CCL2 secretion from HaCaT cells.**

| Cytokine/Chemokine | N | Exposure time /CoCl$_2$ concentration/ High vs. low calcium incubation | AM | 95% CI | P-value |
|---|---|---|---|---|---|
| **IL-6 (pg/ml)** | 39 | 30 minutes | 1.44 | 0.99–1.88 | |
| | 40 | 24 hours | **2.81** | **2.29–3.33** | **0.000** |
| | 40 | 48 hours | **3.08** | **2.56–3.58** | **0.000** |
| | 30 | 0 μM | 1.94 | 1.38–2.51 | |
| | 30 | 100 μM | 2.13 | 1.55–2.70 | 1.000 |
| | 30 | 500 μM | 2.52 | 1.95–3.08 | 1.000 |
| | 29 | 1000 μM | **3.24** | **2.52–3.97** | **0.016** |
| | 60 | Low calcium | 2.79 | 2.36–3.23 | |
| | 59 | High calcium | **2.10** | **1.68–2.53** | **0.024** |
| **CXCL8 (pg/ml)** | 39 | 30 minutes | 4.77 | 4.31–5.23 | |
| | 40 | 24 hours | **6.92** | **6.52–7.31** | **0.000** |
| | 40 | 48 hours | **7.14** | **6.67–7.60** | **0.000** |
| | 30 | 0 μM | 6.02 | 5.42–6.61 | |
| | 30 | 100 μM | 6.26 | 5.82–6.69 | 1.000 |
| | 30 | 500 μM | 6.20 | 5.53–6.87 | 1.000 |
| | 29 | 1000 μM | 6.68 | 5.83–7.53 | 0.873 |
| | 60 | Low calcium | 6.21 | 5.70–6.72 | |
| | 59 | High calcium | 6.36 | 5.99–6.74 | 0.625 |
| **CCL2 (pg/ml)** | 38 | 30 minutes | 2.49 | 1.90–3.08 | |
| | 40 | 24 hours | **4.25** | **3.71–4.79** | **0.000** |
| | 40 | 48 hours | **4.85** | **4.33–5.37** | **0.000** |
| | 30 | 0 μM | 4.52 | 3.64–5.40 | |
| | 30 | 100 μM | 4.30 | 3.58–5.01 | 1.000 |
| | 29 | 500 μM | 3.46 | 2.87–4.04 | 0.202 |
| | 29 | 1000 μM | 3.24 | 2.59–3.89 | 0.065 |
| | 60 | Low calcium | 3.85 | 3.27–4.44 | |
| | 58 | High calcium | 3.92 | 3.50–4.35 | 0.843 |

AM = Arithmetic mean

95% CI = 95% confidence interval of the mean.

ANOVA analyses was used to evaluate whether there are significant differences in secretion from the cells when comparing the controls and respective testing concentration and time of exposure. The difference in secretion between high and low calcium incubation was also evaluated. Significant increase in secretion was seen for time for IL-6, CXCL8 and CCL. Only IL-6 showed an increase between the control and the highest CoCl$_2$ test concentration. The high (1.8 mM) and low (0.15 mM) calcium incubation conditions of the cells only affected IL-6 secretion. [a,b]

[a] All data ln-transformed

[b] n = 120

0.77, 95% CI 0.61–0.98) and *CXCL8* (OR 0.55, 95% CI 0.43–0.71) and increased production for *CCL2* mRNA (OR 1.57, 95% CI 1.21–2.04). High-and-low- calcium incubation conditions showed a significant increase only in *CXCL8* mRNA production (OR = 1.31, 95% CI 1.02–1.70).

## Discussion

HaCaT is widely used in models investigating the effect of irritants or drugs on the skin and the cell line exhibits the major surface markers and displays comparable differentiation as primary isolated human keratinocytes [16, 20]. *In vitro* models using HaCaT cells have been shown to provide a suitable microenvironment in the study of skin sensitization [21]. It has

**Table 2. Linear regression mixed model analyses for dose- and time-trends for IL-6, CXCL8 and $CCL_2$ secretion (relative expression) from HaCaT cells.**

| Cytokine/ Chemokine | Exposure time /$CoCl_2$ concentration/High vs. low calcium incubation | OR | 95% CI |
|---|---|---|---|
| IL-6 | 30 minutes | 1.00 | |
| | 24hours | **3.93** | **2.08–7.43** |
| | 48 hours | **5.11** | **2.70–9.76** |
| | 0 µM | 1.00 | |
| | 100 µM | 1.20 | 0.58–2.49 |
| | 500 µM | 1.77 | 0.85–3.68 |
| | 1000 µM | **3.50** | **1.67–7.32** |
| | Low calcium | 1.00 | |
| | High calcium | **2.00** | **1.19–3.37** |
| CXCL8 | 30 minutes | 1.00 | |
| | 24 hours | **8.49** | **4.58–15.73** |
| | 48 hours | **10.56** | **5.7–19.57** |
| | 0 µM | 1.00 | |
| | 100 µM | 1.27 | 0.63–2.58 |
| | 500 µM | 1.20 | 0.59–2.44 |
| | 1000 µM | 1.85 | 0.91–3.77 |
| | Low calcium | 1.00 | |
| | High calcium | 0.87 | 0.53–1.44 |
| CCL2 | 30 minutes | 1.00 | |
| | 24 hours | **5.98** | **2.88–12.44** |
| | 48 hours | **10.95** | **5.27–22.77** |
| | 0 µM | 1.00 | |
| | 100 µM | 0.80 | 0.35–1.84 |
| | 500 µM | **0.33** | **0.14–0.76** |
| | 1000 µM | **0.26** | **0.11–0.61** |
| | Low calcium | 1.00 | |
| | High calcium | 1.00 | 0.55–1.80 |

OR = Odds ratio 95%

CI = 95% confidence interval.

A linear regression mixed model analyses was used describe the relationship between the different exposure times, cobalt chloride concentrations, and high and low calcium incubation of the cells. Significant increase in secretion was seen for time for IL-6, CXCL8 and CCL2. Cobalt chloride concentration showed an increase in IL-6 but a decrease in CCl2. The high (1.8 mM) and low (0.15 mM) calcium incubation conditions of the cells only affected IL-6 secretion. [a,b]

[a] All data ln-transformed

[b] n = 120

been suggested that normal keratinocytes and transformed cell lines differ in their response to some extent, indicating that interpretations need be made with caution [22]. It is known that cobalt can penetrate the epidermis and thereafter be found both in the cell nucleus and in the cytosol of cells where it can impact DNA repair mechanisms [7, 23]. In the present study, we utilize the culture conditions to model two separate states of differentiation in the HaCaT cell population representing basally located cells of low differentiation level and more

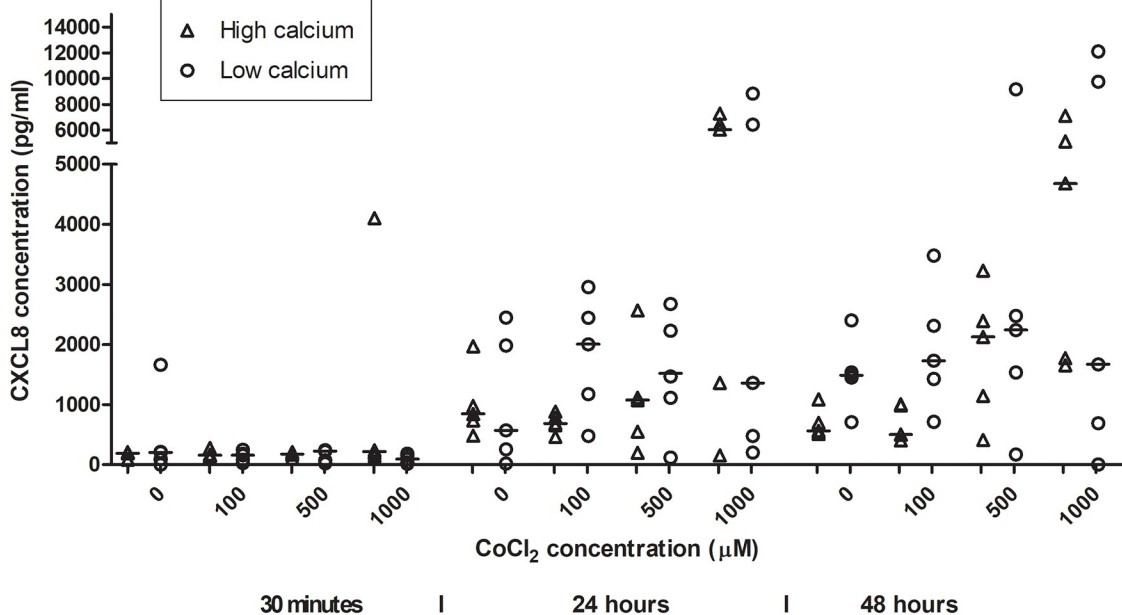

**Fig 4. CXCL8 secretion from HaCaT-cells after cobalt chloride exposure.** HaCaT-cells are exposed to cobalt chloride concentrations 100, 500, 1000 μM and an unexposed control (0 μM) at time points 30 minutes, 24 hours and 48 hours. Increase in CXCL secretion is seen for high (1.8 mM) and (low 0.15 mM) calcium culturing conditions.

differentiated suprabasally located cells in the epidermis. Using extracellular calcium as an inducer for differentiation for HaCaT cells is a well described method [24, 25] and extensively used to study differentiation states of the cells. To our knowledge, this is the first time differentiation in the HaCaT cells has been a parameter in the effect of CoCl$_2$ on keratinocytes.

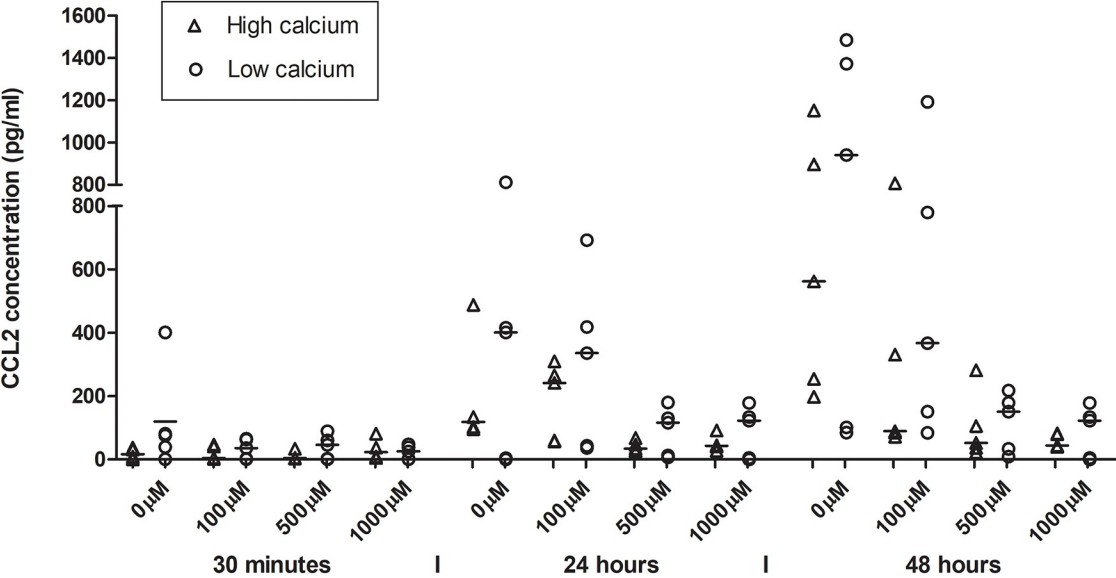

**Fig 5. CCL2 secretion in HaCaT-cells after cobalt chloride exposure.** HaCaT cells are exposed to cobalt chloride concentrations 100, 500 and 1000 μM and an unexposed control (0 μM). Time points are 30 minutes, 24 hours and 48 hours. The CCL2 secretion from the cells increased over time but decreased with increasing cobalt chloride concentration. This was seen for high (1.8 mM) and low (0.15 mM) calcium incubation conditions.

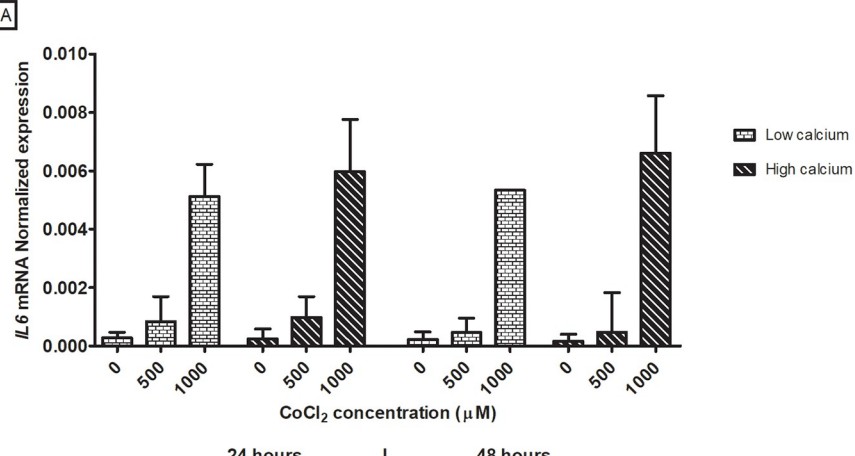

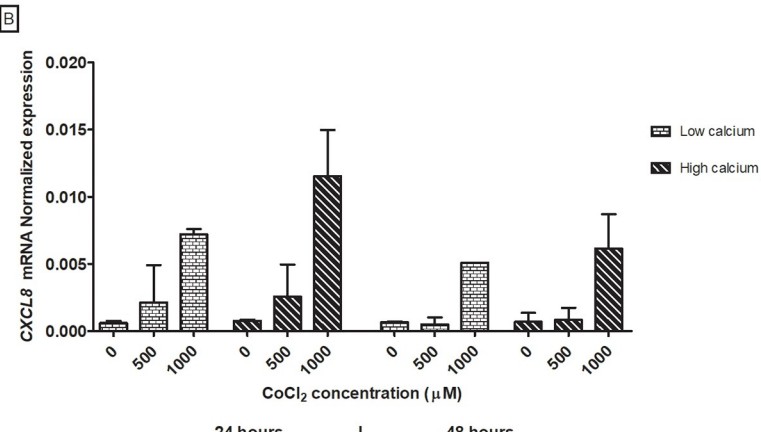

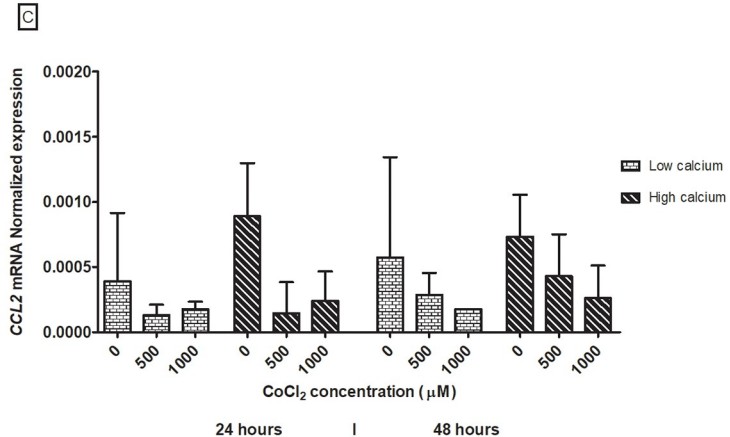

**Fig 6. Gene expression of *IL6* (A), *CXCL8* (B) and *CCL2* (C) in HaCaT cells exposed to cobalt chloride.** The HaCaT cells were exposed for 24 and 48 hours to cobalt chloride concentrations 0 μM, 500 μM and 1000 μM CoCl$_2$. High (1.8 mM) and low (0.15 mM) calcium growth conditions is showed. Results represent median value of the relative mRNA levels and range (min-max). The *IL6*, *CXCL8* and *CCL2* mRNA production was dose- and time-dependent and *CXCL8* mRNA production was affected by the calcium incubation conditions.

**Table 3. ANOVA analyses showing dose- and time-relationships for the relative levels of *IL6* mRNA, *CXCL8* mRNA and *CCL2* mRNA.**

| Genes | N | Exposure time /CoCl$_2$ concentration/High vs. low calcium incubation | Mean | CI 95% | P-value |
|---|---|---|---|---|---|
| *IL6* mRNA | 30 | 24hours | -7.42 | -7.93–(-6.90) | |
| | 35 | 48 hours | -6.88 | -7.33–(-6.44) | 0.111 |
| | 24 | 0 µM | -8.40 | -8.60–(-8.20) | |
| | 24 | 500 µM | **-7.22** | **-7.47–(-6.98)** | **0.000** |
| | 17 | 1000 µM | **-5.20** | **-5.35–(-5.05)** | **0.000** |
| | 34 | Low calcium | -7.03 | -7.55–(-6.52) | |
| | 31 | High calcium | -7.23 | -7.68–(-6.78) | 0.558 |
| *CXCL8* mRNA | 30 | 24hours | -6.83 | -7.20–(-6.47) | |
| | 35 | 48 hours | **-6.03** | **-6.47–(-5.65)** | **0.003** |
| | 24 | 0 µM | -7.25 | -7.34–(-7.15) | |
| | 24 | 500 µM | **-6.64** | **-7.03–(-6.26)** | **0.002** |
| | 17 | 1000 µM | **-4.88** | **-5.03–(-4.72)** | **0.000** |
| | 34 | Low calcium | -6.23 | -6.63–(-5.83) | |
| | 31 | High calcium | -6.59 | -6.98–(-6.20) | 0.199 |
| *CCL2* mRNA | 30 | 24hours | -7.68 | -7.88–(-7.49) | |
| | 35 | 48 hours | **-8.20** | **-8.48–(-7.91)** | **0.005** |
| | 24 | 0 µM | -7.33 | -7.51–(-7.16) | |
| | 24 | 500 µM | **-8.38** | **-8.67–(-8.08)** | **0.000** |
| | 17 | 1000 µM | **-8.25** | **-8.55–(-7.96)** | **0.000** |
| | 34 | Low calcium | -7.84 | -8.08–(-7.60) | |
| | 31 | High calcium | -8.09 | -8.38–(-7.80) | 0.182 |

AM = Arithmetic mean

95% CI = 95% confidence interval of the mean.

ANOVA analysis was performed to determine if there are differences between the controls and the different exposure times, cobalt chloride concentrations, and high (1.8 mM) and low (0.15 mM) calcium incubation. For *IL6*, *CXCL8* and *CCL2* mRNA production a significant increase was seen for increasing testing concentrations. *CXCL8* and *CCL2* mRNA was also affected by the time. [a,b]

[a] All data ln-transformed

[b] n = 65

We found a dose- and time-dependent cytotoxic effect of CoCl$_2$ on cells already detectable after six hours of exposure. These data are supported by reports of the dose-dependent cytotoxic effect of CoCl$_2$ following short exposures (4 hours) and seen at low (100 µM) cobalt concentrations as well higher (1 mM) with the EC50 value for CoCl$_2$ estimated to be about 475 µM [13, 15, 18, 26]. Interestingly, a significant difference in the toxicity response between cells of high and low differentiation status was found for some of the time points and concentrations, suggesting that CoCl$_2$ exposure does affect cell viability among suprabasally located keratinocytes differently compared to basal keratinocytes. As clinically relevant sensitization of skin to irritants have a strong time factor, we investigated the effect of 100 µM CoCl$_2$ (determined as sub-toxic through a series of titration experiments spanning 10 µm to 5 mM CoCl$_2$ over 48 hours) for 14 days. Time did not seem to be a factor in our model as cells remained viable to the same extent throughout the experiment, using this long term exposure.

Investigating the proinflammatory capacity of keratinocytes, we further found a dose- and time-dependent response measured as release of proinflammatory cytokines/chemokines. The release of IL-6 from keratinocytes correlate with increasing CoCl$_2$ concentrations with a

**Table 4. Linear regression mixed model analyses showing dose- and time-relationships for *IL6* mRNA, *CXCL8* mRNA and *CCL2* mRNA.**

| Genes | Exposure time /CoCl$_2$ concentration/High vs. low calcium incubation | OR | CI 95% |
|---|---|---|---|
| ***IL6* mRNA** | 24hours | 1.00 | |
| | 48 hours | **0.77** | **0.61–0.98** |
| | 0 μM | 1.00 | |
| | 500 μM | **3.24** | **1.12–9.34** |
| | 1000 μM | **23.31** | **10.05–54.09** |
| | Low calcium | 1.00 | |
| | High calcium | 1.06 | 0.49–2.26 |
| ***CXCL8* mRNA** | 24hours | 1.00 | |
| | 48 hours | **0.55** | **0.43–0.71** |
| | 0 μM | 1.00 | |
| | 500 μM | **1.83** | **1.36–2.46** |
| | 1000 μM | **9.55** | **6.89–13.24** |
| | Low calcium | 1.00 | |
| | High calcium | **1.31** | **1.02–1.70** |
| ***CCL2* mRNA** | 24hours | 1.00 | |
| | 48 hours | **1.57** | **1.21–2.04** |
| | 0 μM | 1.00 | |
| | 500 μM | **0.35** | **0.26–0.47** |
| | 1000 μM | **0.42** | **0.30–0.58** |
| | Low calcium | 1.00 | |
| | High calcium | 1.27 | 0.98–1.64 |

OR = Odds ratio

95% CI = 95% confidence interval

A linear regression mixed model analysis was performed to describe the relationship between the different exposure times, cobalt chloride concentrations, and high (1.8 mM) and low (0.15 mM) calcium incubation. *IL6*, *CXCL8* and *CCL2* mRNA production in the HaCaT cells was affected by time and cobalt chloride concentration. Only *CXCL8* mRNA was affected by calcium incubations for the cells. [a,b].

[a] All data ln-transformed

[b] n = 65

corresponding decrease in CCL2 levels, whereas IL-6, CXCL8, and CCL2 are all showing a correlation with increases over time. The inflammatory response in the HaCaT cells seen as an increase in IL-6 and a decrease in CCL2 secretion at the highest test concentration 1000 μM is interesting as the viability was decreased by near 50%. Increasing time also strongly affected the cytokine/chemokine secretion in the cells. These findings are interesting in a clinical perspective showing an ongoing inflammatory response in the cells despite the metabolic disruption. The dose-dependency is reflected in the mRNA expression and the data indicating that the cells are activated in an acute response to high cobalt exposure. The fact that the mRNA data does not mirror the time-dependent effect, which may reflect that the exposed HaCaT cells have already peaked in mRNA expression of *IL6* and *CXCL8*, which could explain the increases found in protein levels. CCL2 expression increased with time, which may indicate the biological function as a recruitment molecule for monocytes to enter the tissue during inflammatory events in a delayed reaction.

Since our model allows for the investigation of keratinocytes at different states of differentiation, we compared the effects of CoCl$_2$ on cells of the two different states of differentiation.

Interestingly, the more differentiated keratinocytes representing suprabasally located cells responded stronger to $CoCl_2$ exposure as compared to cells of low differentiation level concerning IL-6 secretion and *CXCL8* mRNA production. In addition to the $CoCl_2$ provoked inflammatory response in a dose- and time-dependent manner, a considerable basal production of IL-6 and CXCL8 was found indicating constitutive signaling capacity in skin keratinocytes as proposed by others [17].

IL-6 in the skin is primarily produced by keratinocytes [27], however, the function of IL-6 in the skin is not fully understood but has been associated with the thickening of stratum corneum [28] and increasing the number of epithelial cell layers [29]. IL-6 is elevated in skin inflammation such as psoriasis, atopic dermatitis, and allergic contact dermatitis, but also associated with skin healing [30–33]. The relevance of our results showing a dose- and time-dependent increase in IL-6 levels as a response to $CoCl_2$ exposure is supported by the fact that IL-6 plays an important part in several events during inflammatory states of the skin and possibly physical protection against harmful stimuli by forming thicker skin.

CXCL8 is a proinflammatory chemokine and plays a role in the early accumulation of leukocytes, especially neutrophils, in the skin and is, therefore, an important part of the delayed type of hypersensitivity reaction, i.e., allergic dermatitis [34]. CXCL8 has been shown to be linked to several inflammatory states including psoriasis [35] and being involved in wound healing [36]. The main role of CXCL8 in inflammation is to attract neutrophils [37], but also have a role in chemotactic migration and activation of monocytes and lymphocytes [38]. In our study, we observed an increase in CXCL8 concentrations at 24 and 48 hours. We decreased in *CXCL8* mRNA expression at 48 hours after exposure to $CoCl_2$, which could be supported by the fact that CXCL8 is secreted in the initial phase of acute inflammation and the need to recruit neutrophils.

In addition to increasing levels of IL-6 production following exposure to $CoCl_2$, a marked decline in CCL2 secretion and *CCL2* mRNA expression was detected as a response to increasing $CoCl_2$ concentrations. Similar results have been previously described in cell models where $CoCl_2$ is used as an hypoxia mimicking agent, and downregulation of CCL2 expression has been described in ovarian tumor cells [39] and monocytes and fibroblasts [40]. CCL2 is one of the key chemokines that regulate migration and infiltration of monocytes, but also lymphocytes [41, 42] and natural killer cells to the site of inflammation [43] and is also suggested to be involved in Th0 towards Th2 polarization [44]. CCL2 is seen in several inflammatory diseases, including psoriasis and atopic contact dermatitis [45, 46]. In nickel induced occupational dermatitis, the expression of CCL2 declines gradually 72–96 hours after nickel application, correlating with our findings using cobalt [46].

In our model using HaCaT cells to investigate the cytotoxic and inflammatory response in different differentiation status of the cells could be a limitation since HaCaT cells can differ in their response and primary keratinocytes may have been an alternative to use in the model. But since the response in the cells depending on differentiation state was a major parameter in the experiment, HaCaT cells was chosen due to the well established method of using calcium concentrations during culturing to be able to perform identical and simultaneously exposure experiments on the cells in the two differentiation states. Investigating the differentiation states of keratinocytes is a context of $CoCl_2$ dose- and time response is interesting in a clinical perspective as penetration of the metal depends on concentration of the metal and duration of the exposure. Many humans are exposed to cobalt in their daily life from contact with cobalt contacting products in their daily life and in industrial settings i.e, during hard metal production and at which exposure could be over a long period of time.

## Conclusions

Using our cell model, we have described that $CoCl_2$ exposure is cytotoxic to HaCaT cells as well as affecting a potent proinflammatory response important for the recruitment of immunocompetent cells and central for skin conditions such as dermatitis and other inflammatory skin conditions. Our protein and mRNA data reflect a classical inflammatory event in the dermis where the stimulation triggers an initial acute response featuring IL-6 and CXCL8, affecting the epithelium and facilitating recruitment of neutrophils with a delayed signal in CCL2 mediating recruitment of monocytes and resolution of inflammation.

The HaCaT cells respond to $CoCl_2$ exposure in a dose- and time-dependent manner and that cell viability decreases with increasing time and concentrations. The secreted cytokines are involved in the recruitment of leukocytes, especially neutrophils, macrophages, natural killer cells, but also inhibition of neutrophils, cells which upon massive infiltration are central for the development of skin maladies. Our findings further suggest that the effect of cobalt chloride on cell toxicity occurs throughout the living epidermis.

## Author Contributions

**Conceptualization:** Magnus Lindberg, Eva Särndahl, Håkan Westberg, Alexander Persson.

**Formal analysis:** Maria Klasson, Ing-Liss Bryngelsson, Kedeye Tuerxun, Alexander Persson.

**Funding acquisition:** Maria Klasson, Magnus Lindberg, Eva Särndahl.

**Investigation:** Maria Klasson, Kedeye Tuerxun.

**Methodology:** Maria Klasson, Magnus Lindberg, Eva Särndahl, Alexander Persson.

**Project administration:** Maria Klasson, Alexander Persson.

**Resources:** Magnus Lindberg.

**Supervision:** Magnus Lindberg, Eva Särndahl, Håkan Westberg, Alexander Persson.

**Validation:** Maria Klasson, Kedeye Tuerxun, Alexander Persson.

**Visualization:** Maria Klasson, Magnus Lindberg, Alexander Persson.

**Writing – original draft:** Maria Klasson, Alexander Persson.

**Writing – review & editing:** Maria Klasson, Magnus Lindberg, Eva Särndahl, Håkan Westberg, Ing-Liss Bryngelsson, Kedeye Tuerxun, Alexander Persson.

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
