## [Decision Letter · Decision Letter 0]

8 Mar 2021

PONE-D-21-02649

Dose- and time-dependent changes in viability and IL-6, CXCL8 and CCL2 production by HaCaT-cells exposed to cobalt. Effects of high and low calcium growth conditions.

PLOS ONE

Dear Dr. Klasson,

Thank you for submitting your manuscript to PLOS ONE. After careful consideration, we feel that it has merit but does not fully meet PLOS ONE’s publication criteria as it currently stands. Therefore, we invite you to submit a revised version of the manuscript that addresses the points raised during the review process.

We look forward to receiving your revised manuscript.

Kind regards,

Yi Cao

Academic Editor

PLOS ONE

Journal Requirements:

2)  Please provide additional information about each of the cell lines used in this work, including any quality control testing procedures (authentication, characterisation, and mycoplasma testing). For more information, please see http://journals.plos.org/plosone/s/submission-guidelines#loc-cell-lines.

3) Please ensure you have discussed any potential limitations of your study in the Discussion.

Reviewers' comments:

Reviewer's Responses to Questions

**Comments to the Author**

1. Is the manuscript technically sound, and do the data support the conclusions?

Reviewer #1: Yes

Reviewer #2: Partly

2. Has the statistical analysis been performed appropriately and rigorously? 

Reviewer #1: Yes

Reviewer #2: No

3. Have the authors made all data underlying the findings in their manuscript fully available?

Reviewer #1: Yes

Reviewer #2: Yes

4. Is the manuscript presented in an intelligible fashion and written in standard English?

Reviewer #1: Yes

Reviewer #2: Yes

5. Review Comments to the Author

Reviewer #1: The manuscript is partially original in your findings once there are several manuscripts about cobalt citotoxicity in HaCaT cells. The authors performed the assays with HaCaT cells, an imortalized cell line. Why the authors performed with this cells line and not with primary keratinocytes?

Reviewer #2: The authors studied the dose- and time-dependent changes in viability and IL-6, CXCL8 and CCL2

production by HaCaT-cells exposed to cobalt.

This is an interesting study because the molecular description of the proinflammatory response on keratinocytes is innovative.

The article is well written and the methods are well described.

Several points caught our attention :

1/ the processing of the test results is done with unusual methods

2/ the concentrations at which effects are detected are very high.

Major remarks

1/ cytokine production

The experimental data are given in fig. 3, 4 and 5.

The dispersion of the results is very astonishing. A very large variation between the experimental results for one dose and one time is observed. For example, for CXCL8 after exposure to 500µM Co during 48h, the concentration between experiments varies from 0 to 10,000 pg/mL. Such a dispersion is not expected for identical experiments....

These results should first be analyzed using standard statistical tools using an ANOVA test to validate whether the differences between the control (0µM) and the other concentrations are statistically significant.

2/ RT-PCR

The authors do not explain why they used a calibration curve performed with blood mononuclear cells exposed to LPS while analyzing the response of HaCaT cells. Usually, RT-PCR results are processed using relative quantification calculated using the ΔΔCt method normalized to the reference genes. Then, relative normalized expression, change in expression relative to control samples, and p value are calculated.

Examples on HaCaT cells : PMID: 32493482, PMID: 31216667, PMID: 33192123

This universal method should be used for processing RT-PCR data.

3/ Since conventional methods of data analysis are not used, it is difficult to appreciate the differences induced by cobalt. Nevertheless, the authors describe changes in cytokine expression and mRNA modulation at 500 and 1000µM. Drawing conclusions on the inflammatory response at a Co concentration of 1000 µm seems questionable, especially after 48h exposure, as in this case cell viability is reduced by more than 50% (Fig. 1). This point should be discussed.

4/ the authors cannot write in the discussion section that “The release of IL-6 and CXCL8 from keratinocytes correlate with increasing CoCl2 concentrations” and “In addition to increasing levels of IL-6 and CXCL8 production following exposure to CoCl2” when they write in the results section “non-significant increases of CXCL8 were noted for the different CoCl2 concentrations”. Consistency between the different statements must be checked.

6. PLOS authors have the option to publish the peer review history of their article (what does this mean?). If published, this will include your full peer review and any attached files.

Reviewer #1: No

Reviewer #2: No

---

## [Author Response · Author response to Decision Letter 0]

7 May 2021

Dear editor and reviewers,

Thank you for the comments and the opportunity of re-consideration of our manuscript entitled “Dose- and time-dependent changes in viability and IL-6, CXCL8 and CCL2 production by HaCaT-cells exposed to cobalt. Effects of high and low calcium growth conditions.”

We have considered the comments of the reviewers, several of which amended into the manuscript as signific improvements, and commented all of them in our detailed response. In particular for the major comments we have: 

- Validated that the differences in cytokine/chemokine secretion and gene expression between the control (0µM) and the other concentrations are statistically significant by performing ANOVA (table 2 and 4). We also validate the difference in exposure time and between high and low calcium.

- Commented on the wide dispersion of produced cytokines/chemokines in response to CoCl2 exposure as well as the inflammatory response at the highest test concentration and reduction in viability of the cells. 

-Explained our rationale for using relative standard curve method when interpreting the PCR data.

Journal Requirements:

We have checked the templates and made the few adjustments to meet the journal requirements.

2) Requirements: Please provide additional information about each of the cell lines used in this work, including any quality control testing procedures (authentication, characterisation, and mycoplasma testing). For more information, please see http://journals.plos.org/plosone/s/submission-guidelines#loc-cell-lines.

The HaCaT cell line used in the experiments are tested for contamination of mycoplasma as frequent routine control using the antibiotics free cell culture medium from 14 days constitutive culture using the MycoAlertTM Mycoplasma Detection Kit from Lonza, Basel, Switzerland. 

Method: p. 6, 1st para, line 101-104.

3) Please ensure you have discussed any potential limitations of your study in the Discussion.

We have discussed the limitation in using HaCaT cells as suggested.

Discussion: p. 28, 2nd para, line 454-465.

Comments to the Author:

Reviewer #1

The manuscript is partially original in your findings once there are several manuscripts about cobalt citotoxicity in HaCaT cells. The authors performed the assays with HaCaT cells, an imortalized cell line. Why the authors performed with this cells line and not with primary keratinocytes?

Being a cell line, we are aware of the potential differences in response between HaCaT compared to primary keratinocytes. In order to evaluate the effect of cobalt on keratinocytes the choice of HaCaT was rather uncontroversial due to the vast body of available knowledge regarding their functionality to evaluate the current data in a relevant context. Since the response in the cells depending on differentiation state of the cells was a major parameter in the current manuscript and that this model is by far the most well described approach to model keratinocyte differentiation, HaCaT was a natural choice. Important was also to be able to perform identical and simultaneously exposure experiments on the cells in the two differentiation states and since the experiments required much cells, the HaCaT model was most feasible to conduct the research.

To our knowledge, this is the first time the differentiation states of the cells has been a parameter when investigating CoCl2 cytotoxicity and inflammatory response in HaCaT. 

Discussion: p. 25, 1st para, line 377-328 and 385-387. p. 28, 2nd para, line 454-465.

Reviewer #2

The authors studied the dose- and time-dependent changes in viability and IL-6, CXCL8 and CCL2

production by HaCaT-cells exposed to cobalt.

This is an interesting study because the molecular description of the proinflammatory response on keratinocytes is innovative.

The article is well written and the methods are well described.

Several points caught our attention :

1/ the processing of the test results is done with unusual methods

2/ the concentrations at which effects are detected are very high.

1/ Using a linear mixed model to determine relationships may seem bold and unorthodox for describing biomedical effects. However, since we have repeated measurements in the experiments, we used the linear mixed model to be able to evaluate the effect between the experiments as well as within the experiments. A major benefit of using this model is, since the effects are small, to be able to detect the effect each parameter has on the outcome. In an occupational exposure situation for example it is not the acute effects from the exposure that we can measure. In those contexts it is important to take all possible parameters in count that can negatively affect the health. 

Another major advantage of using the linear mixed model is that the results presented in odds ratio offer understanding of the magnitude of the effects found to be significantly different. By that, different mechanisms can be weighed providing something that we have not seen before in similar studies. 

2/ We find it interesting to include a seemingly high exposure concentration as we saw an inflammatory response despite that the cells viability are affected.

Major remarks

1/ cytokine production

The experimental data are given in fig. 3, 4 and 5.

The dispersion of the results is very astonishing. A very large variation between the experimental results for one dose and one time is observed. For example, for CXCL8 after exposure to 500µM Co during 48h, the concentration between experiments varies from 0 to 10,000 pg/mL. Such a dispersion is not expected for identical experiments....

These results should first be analyzed using standard statistical tools using an ANOVA test to validate whether the differences between the control (0µM) and the other concentrations are statistically significant.

The dispersion in the cytokine/chemokine secretion from the cells are large between experiments but we have chosen to include all available experimental data in the figures. Despite the dispersion, differences are identified which rather strengthen the findings. 

We have added ANOVA analyses on the cytokine/chemokine and mRNA data for validation of if the differences between the control and the other test concentrations are statistically significant. We also included time and calcium incubation.

Methods: p. 9, 2nd para, line 170-174 and p.10, 2nd para, line 203-204.

Results: Cytokines/chemokines: p.13, 1st para, line 234-239, 2nd para, line 245-247 and 3rd para line 252-254. Table 1, p.15-16. 

Gene expression: p.19, 2nd para, line 319-326. Table 3, p.21-22. 

2/ RT-PCR

The authors do not explain why they used a calibration curve performed with blood mononuclear cells exposed to LPS while analyzing the response of HaCaT cells. Usually, RT-PCR results are processed using relative quantification calculated using the ΔΔCt method normalized to the reference genes. Then, relative normalized expression, change in expression relative to control samples, and p value are calculated.

Examples on HaCaT cells : PMID: 32493482, PMID: 31216667, PMID: 33192123

This universal method should be used for processing RT-PCR data.

As the reviewer points out, ΔΔCt method is a traditionally used way of calculating relative differences in expression between two samples. The relative standard curve method is in many ways similar to this method, but has a few advantages mainly related to stability and accuracy in interpretation of data since the data is derived from an actual comparison rather than depend on cycle analysis. The LPS-stimulated PBMCs should be considered a stock pool of a high amount of RNA for the genes of interest, and is included as a calibrator sample in the analysis, hence the actual source of the material is not affecting the interpretation of the data.

The advantages with relative standard curve method over the ΔΔCt method is that it requires the least amount of validation because the PCR efficiencies of the target and endogenous control do not have to be equivalent (which is a fundamental assumption using the ΔΔCt method where all primer pairs require identical efficiency for the ΔΔCt calculation to be correct). Hence, extensive optimization is a fundamental requirement if the ΔΔCt method is to be used in the data analysis. The relative standard curve method gives highly accurate quantitative results because unknown sample quantitative values are interpolated from the standard curve. However, this requires more reagents (to run standard curves on each plate). This relative quantification method is recommended when testing low numbers of targets and small numbers of samples and the interest includes identification of very discrete expression changes.

The relative standard curve method is described in a number of available qPCR handbooks:

https://assets.thermofisher.com/TFS-Assets/LSG/manuals/cms_042380.pdf

https://www.sigmaaldrich.com/content/dam/sigma-aldrich/docs/Sigma-Aldrich/General_Information/1/pcr-technologies-guide.pdf

https://www.gene-quantification.de/national-measurement-system-qpcr-guide.pdf

The relative standard curve method is further described in the literature (a few examples):

https://pubmed.ncbi.nlm.nih.gov/16972087/

https://pubmed.ncbi.nlm.nih.gov/15780134/

3/ Since conventional methods of data analysis are not used, it is difficult to appreciate the differences induced by cobalt. Nevertheless, the authors describe changes in cytokine expression and mRNA modulation at 500 and 1000µM. Drawing conclusions on the inflammatory response at a Co concentration of 1000 µm seems questionable, especially after 48h exposure, as in this case cell viability is reduced by more than 50% (Fig. 1). This point should be discussed.

We have discussed this point as suggested.

Discussion: p. 26, 2nd para, line 405-410.

4/ the authors cannot write in the discussion section that “The release of IL-6 and CXCL8 from keratinocytes correlate with increasing CoCl2 concentrations” and “In addition to increasing levels of IL-6 and CXCL8 production following exposure to CoCl2” when they write in the results section “non-significant increases of CXCL8 were noted for the different CoCl2 concentrations”. Consistency between the different statements must be checked.

We have changed the sentences as suggested for clarification and consistency in the statements.

Sincerely,

Maria Klasson

---

## [Editor Report · Decision Letter 1]

11 May 2021

Dose- and time-dependent changes in viability and IL-6, CXCL8 and CCL2 production by HaCaT-cells exposed to cobalt. Effects of high and low calcium growth conditions.

PONE-D-21-02649R1

Dear Dr. Klasson,

We’re pleased to inform you that your manuscript has been judged scientifically suitable for publication and will be formally accepted for publication once it meets all outstanding technical requirements.

Kind regards,

Yi Cao

Academic Editor

PLOS ONE
---

## [Editor Report · Acceptance letter]

25 May 2021

PONE-D-21-02649R1 

Dose- and time-dependent changes in viability and IL-6, CXCL8 and CCL2 production by HaCaT-cells exposed to cobalt. Effects of high and low calcium growth conditions. 

Dear Dr. Klasson:

I'm pleased to inform you that your manuscript has been deemed suitable for publication in PLOS ONE. Congratulations! Your manuscript is now with our production department. 

Kind regards, 

on behalf of

Dr. Yi Cao 

Academic Editor

PLOS ONE